# Plasma protein profiling predicts cancer in patients with non-specific symptoms

Fredrika Wannberg [1,11], María Bueno Álvez [2,11], Alvida Qvick [3,11], Tamas Pongracz [1], Katherina Aguilera[1], Emma Adolfsson [4], Louise Essehorn[5], Max Gordon [6], Mathias Uhlén [2], Gisela Helenius[3,7], Viktoria Hjalmar[1,8], Mikael Åberg [9,11], Axel Rosell [1,10,11] & Charlotte Thålin [1,11] ✉

Cancer detection is challenging, especially in patients with diffuse symptoms that overlap with non-malignant conditions. Here we show that plasma protein profiling can identify cancer among patients with non-specific symptoms. Using proximity extension assay-based proteomics of 1463 plasma proteins from 456 patients presenting with non-specific symptoms sampled prior to cancer diagnostic work-up and diagnosis, we identify 29 proteins associated with new cancer diagnoses. We develop a model able to stratify 160 cancer cases and 296 non-cancer cases with an area under the curve of 0.80, maintaining performance (0.82) in an independent replication cohort of 238 patients. The model also distinguishes cancer from autoimmune, inflammatory and infectious diseases. Designed as a triage tool, our model based on a blood test could help prioritize patients at higher cancer risk for rapid and highly sensitive diagnostic modalities such as positron emission tomography–computed tomography. These findings emphasize the potential of blood proteome profiling to support timely diagnosis and transform clinical medicine.

Early detection of cancer is key to reduce cancer-related mortality and morbidity[1]. However, early diagnosis is challenging in patients presenting with non-specific symptoms[2] as these patients are not entitled to one of the organ-specific cancer diagnostic pathways. Cancer screening programs furthermore primarily target a limited number of cancer types, such as breast, cervical and colorectal cancer[3,4]. To avoid diagnostic delay in this patient group, several countries have therefore initiated fast-track diagnostic pathways for patients with non-specific symptoms suggestive of an underlying cancer[2,5–7]. Referral of all patients

with non-specific symptoms to fast-track diagnostic pathways would, however, strain the healthcare system and expose patients to potentially harmful and unnecessary investigations. Minimally invasive and easily accessible blood-based biomarkers discriminating between patients at high risk of cancer and those without cancer could aid in selecting patients that would benefit the most from an accelerated and targeted cancer diagnostic work-up and possibly enable earlier diagnosis.

Pan-cancer blood biomarkers have been the focus of extensive research[8–20]. While a handful of pan-cancer blood biomarkers have

[1]Department of Clinical Sciences, Danderyd Hospital, Division of Internal Medicine, Karolinska Institutet, Stockholm, Sweden. [2]Science for Life Laboratory, Department of Protein Science, KTH Royal Institute of Technology, Stockholm, Sweden. [3]Clinical Research Center, Faculty of Medicine and Health, Örebro University, Örebro, Sweden. [4]Department of Obstetrics and Gynecology, Faculty of Medicine and Health, Örebro University, Örebro, Sweden. [5]Division of Internal Medicine, Danderyd Hospital, Stockholm, Sweden. [6]Department of Clinical Sciences, Danderyd Hospital, Division of Orthopedics, Karolinska Institutet, Stockholm, Sweden. [7]ATMP-center, Skåne University Hospital, Lund, Sweden. [8]Division of Specialist Medical Care, Diagnostic center, Danderyd Hospital, Stockholm, Sweden. [9]Department of Medical Sciences, Clinical Chemistry and SciLifeLab Affinity Proteomics, Uppsala University, Uppsala, Sweden. [10]Center for Hematology and Regenerative Medicine (HERM), NEO, Department of Medicine, Huddinge, Karolinska Institutet, Stockholm, Sweden. [11]These authors contributed equally: Fredrika Wannberg, María Bueno Álvez, Alvida Qvick, Mikael Åberg, Axel Rosell, Charlotte Thålin. ✉e-mail: charlotte.thalin@ki.se

been evaluated in patient cohorts with non-specific symptoms prior to cancer diagnosis, such as soluble urokinase plasminogen activator receptor[21], cfDNA methylation patterns[22] and markers of neutrophil extracellular traps[23], the vast majority of pan-cancer blood biomarkers are first explored in patient cohorts with known cancer[24] or are limited by the use of healthy control groups[8,9,11–18,20,25] which do not reflect the clinical setting in which pan-cancer blood biomarkers are intended to be implemented. This lack of real-world settings during the discovery phase of cancer biomarkers likely contributes to low accuracy in detecting early-stage cancers and hinders their successful integration into clinical practice[24].

Given the vast heterogeneity of cancer, identifying unique pan-cancer blood biomarkers is challenging. Therefore, integrating multiple biomarkers or biomarker candidates into a combined cancer test is an attractive strategy to enhance accuracy. Plasma proteomics is a rapidly expanding field, and recent technological advancements enable quantification of thousands of proteins using minute amounts of plasma in large cohorts[26].

Here, we used the proximity extension assay-based proteomics analysis of 1463 proteins to explore the proteomic signature in blood samples collected from 456 patients presenting with non-specific symptoms prior to extensive cancer diagnostic workup. We identified a core set of proteins associated with a new cancer diagnosis. This protein signature was able to discriminate between patients later diagnosed with cancer and patients diagnosed with non-malignant autoimmune and inflammatory disorders, which are disease entities especially difficult to distinguish from malignancies[27]. We confirmed the ability of the protein signature to discriminate between cancer and non-cancer in an independent cohort of 238 patients presenting with similar non-specific symptoms and referred to a comparable fast-track diagnostic pathway at a different domestic hospital. Our findings indicate that cancer-specific signatures, identified through next-generation protein profiling, can be leveraged in a blood test to detect patients with non-specific symptoms who are at elevated risk of an underlying malignancy. Rather than replacing highly sensitive diagnostic work-ups, such a test could help distinguish cancer from non-malignant inflammatory conditions in heterogeneous, real-world populations, and enable clinicians to prioritize those most likely to benefit from timely, resource-intensive investigations, including advanced imaging and biopsies where appropriate.

## Results

### Discovery cohort

For the discovery of cancer-specific proteomic signatures in patients presenting with non-specific symptoms, we characterized the plasma proteome of 456 patients (55% female, median age 71.0 [IQR 60–78]) referred to the fast-track multidisciplinary cancer diagnostic pathway at Danderyd Hospital, Stockholm, Sweden (the MEDECA cohort; NCT06355245[23]). Referrals to this pathway were based on non-specific symptoms such as general malaise, extreme fatigue, reduced appetite, unintentional weight loss, prolonged fever, unexplained pain, pathological laboratory values, increased number of healthcare contacts, or radiological findings suggestive of metastasis without an apparent primary tumor. All study patients underwent a standardized and extensive cancer diagnostic work up (i.e., fast-track pathway), including an expanded panel of biochemical analyses, diagnostic tissue biopsies and imaging. Among these 456 patients, 160 were diagnosed with cancer during the follow-up period of six months (Supplementary Table S1, Fig. 1). The most common cancer types were hematologic malignancies (28%), pancreas, gall bladder and bile duct adenocarcinomas (11%) and lung adenocarcinomas (8%) (Supplementary Table S2). Metastatic disease was diagnosed in 102 out of 115 patients with solid tumors. Fifty-five of the patients not diagnosed with cancer were diagnosed with a non-malignant autoimmune disease, 19 with an inflammatory disease, and 32 with an infectious disease. One hundred

and thirty-eight patients received no diagnosis, and 52 patients received a non-inflammatory, non-autoimmune or non-infectious diagnosis (Supplementary Table S3). Patients categorized as "no diagnosis" were either given a functional or a non-somatic diagnosis, were diagnosed with a benign tumor condition that did not explain their symptoms or had symptoms and/or pathological findings that remained unexplained following the diagnostic work-up.

### Replication cohort

To replicate the plasma proteomic signature and validate the performance of the classification model in an independent patient cohort, we characterized the plasma proteome of 238 patients (50% female, median age 72 [IQR 61–78]) admitted to a similar fast-track multidisciplinary cancer diagnostic pathway for patients presenting with non-specific symptoms at Örebro University Hospital, Örebro, Sweden (the ALLVOS cohort). Referrals to this pathway were based on similar criteria as above with the exception of a radiological sign of malignancy without an apparent primary site. All study patients underwent a standardized and extensive cancer diagnostic work up as described above. A total of 35 patients were diagnosed with cancer during the six month follow-up period in this cohort (Supplementary Table S1, Fig. 1). The most common cancer types were hematologic malignancies (20%), squamous cell carcinoma in lung (12%) and neuroendocrine tumors (9%) (Supplementary Table S2). Metastatic disease was diagnosed in 12 out of 26 patients with solid tumors. Twenty-six of the patients not diagnosed with cancer were diagnosed with an autoimmune disease, 18 with an inflammatory disease, and 23 with an infectious disease. Eighty-three patients received no diagnosis, and 53 patients received a non- inflammatory, non-autoimmune or non-infectious diagnosis (Supplementary Table S4).

### Differential plasma protein expression patterns characterize patients later diagnosed with cancer as compared to non-cancer in patients with non-specific symptoms of cancer

Samples collected before diagnostic work-up and diagnosis from a total of 694 patients from the two cohorts were analyzed, where all but one sample passed quality control (from the discovery cohort, Fig. 1). Additionally, we observed a high inter-panel correlation for assays used as technical controls, with correlation coefficients of 0.97–0.99 for IL6, 0.97–0.99 for CXCL8 and 0.95–0.96 for TNF (Supplementary Fig. S1).

To identify proteins significantly associated with a new cancer diagnosis, we conducted a differential expression analysis by performing a cancer to non-cancer comparison within the discovery cohort (Fig. 2A). In total, 28 proteins were differentially expressed between patients later diagnosed with cancer and non-cancer patients with an adjusted $p$-value < 0.05 and log fold change (logFC) > 0.5. Strikingly, among these, 22 proteins were upregulated in patients later diagnosed with cancer also in the replication cohort (Fig. 2A). Logistic regression models adjusted for age and sex confirmed that elevated levels of these 22 proteins were significantly associated with increased cancer risk, reflected by significant odds ratios (ORs) per 1 NPX increase greater than 1 (Fig. 2B, Supplementary Table S5). Several of these 22 proteins have been proposed to be cancer-related according to the Human Protein Atlas (v24.proteinatlas.org) (Fig. 2B), but only four are reported as secreted and one as tissue enriched. The plasma protein levels for the seven proteins with the lowest adjusted $p$-value are shown in Fig. 2C and were upregulated in patients later diagnosed with cancer relative to non-cancer patients in both the discovery cohort and the replication cohort.

To further explore the separation of patients later diagnosed with cancer and non-cancer, we conducted principal component analysis (PCA) based on the proteins differentiated in both cohorts ($n = 22$). The first two principal components (PC1 and PC2) were examined to understand the underlying structure of the data among

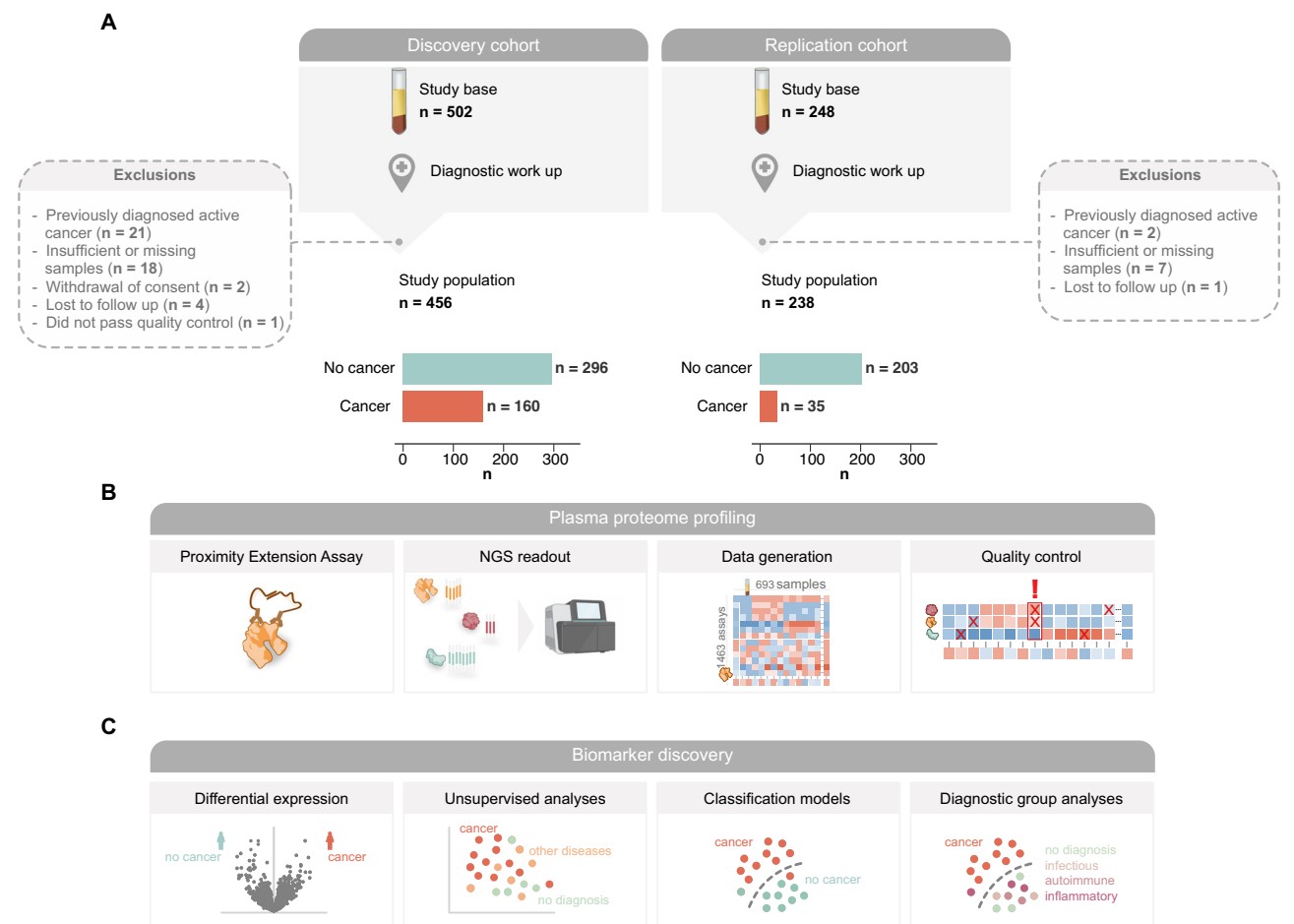

**Fig. 1 | Study populations and biomarker discovery workflow.** Overview of **A** study participants in the discovery and replication cohorts, **B** plasma proteome profiling workflow and **C** biomarker discovery workflow. NGS Next Generation Sequencing.

three distinct groups: patients later diagnosed with cancer, non-malignant diagnoses (including autoimmune, infectious, and inflammatory diagnoses, among others) (Supplementary Tables S3, S4), and patients receiving no diagnosis (Fig. 3A). Most of the variation was explained by PC1, accounting for 65% and 48% of the variation in the discovery and the replication cohorts, respectively. Notably, we observed a left-right separation between the groups along PC1, from no diagnosis to other non-malignant diagnoses to cancer diagnoses. The loading scores for PC1 for the separate proteins were similar across both cohorts (Fig. 3B) and the Normal Protein eXpression (NPX) values were higher for several of these proteins in cancer compared to both no diagnosis and other diagnoses across both cohorts (Fig. 3C). These findings illustrate that the expression of these proteins exhibit variation across disease entities, emphasizing their potential in diagnostics.

**Development and validation of a proteomics-derived multivariate pan-cancer classification model for patients with non-specific symptoms of cancer**

Starting with the discovery cohort, we developed a multivariate penalized logistic regression model to predict cancer in patients with non-specific symptoms based on the proteomics data. Patients were randomly divided (70/30) into a training and test set (Fig. 4A). Within the training set, we applied three initial filtering criteria to reduce the numbers of features prior to model fitting, where a protein needed to meet at least one: (i) a mean NPX difference greater than 1, (ii) an adjusted $p$-value $< 1E{-}6$ from a $t$-test, or (iii) an area under the receiver operating curve (AUC) $> 0.7$ for individual proteins. This selection process

identified 29 proteins (Fig. 4B), which we combined into a feature set used as input to a penalized logistic regression model (lasso) (Fig. 4A, bottom).

With this approach, we identified several new candidate biomarkers for pan-cancer detection (Fig. 4C). All proteins contributed positively, consistent with only upregulated proteins in the differential expression analyses. The developed model performed with an AUC of 0.80 in the discovery cohort, with a similar performance with an AUC of 0.82 when applied to the replication cohort (Fig. 4D). The probability distributions for both cohorts are summarized in Fig. 4E. The results were robust across 50 different seeds (Supplementary Fig. S2), with negligible multicollinearity between the proteins (Supplementary Fig. S3).

To further investigate model performance, we divided patients not diagnosed with cancer in the discovery and replication cohorts into subgroups representing clinical entities as follows: no diagnosis ($n = 141$), infectious disease ($n = 32$), autoimmune disease ($n = 55$), and inflammatory disease ($n = 19$) for the discovery cohort and no diagnosis ($n = 83$), infectious disease ($n = 23$), autoimmune disease ($n = 26$) and inflammatory disease ($n = 18$) for the replication cohort. Patients receiving a diagnosis outside these disease entities were excluded from this analysis due to heterogeneity in this group. We observed a gradient from moderate (AUC 0.67) to high (AUC 0.86) discrimination between cancer and inflammatory disease, cancer and autoimmune disorders, cancer and infectious disorders and cancer and no diagnosis in the discovery cohort, and high discrimination between cancer and all these disease entities in the replication cohort (AUC ranging from 0.82 to 0.84) (Fig. 5). Although this analysis is limited by the small

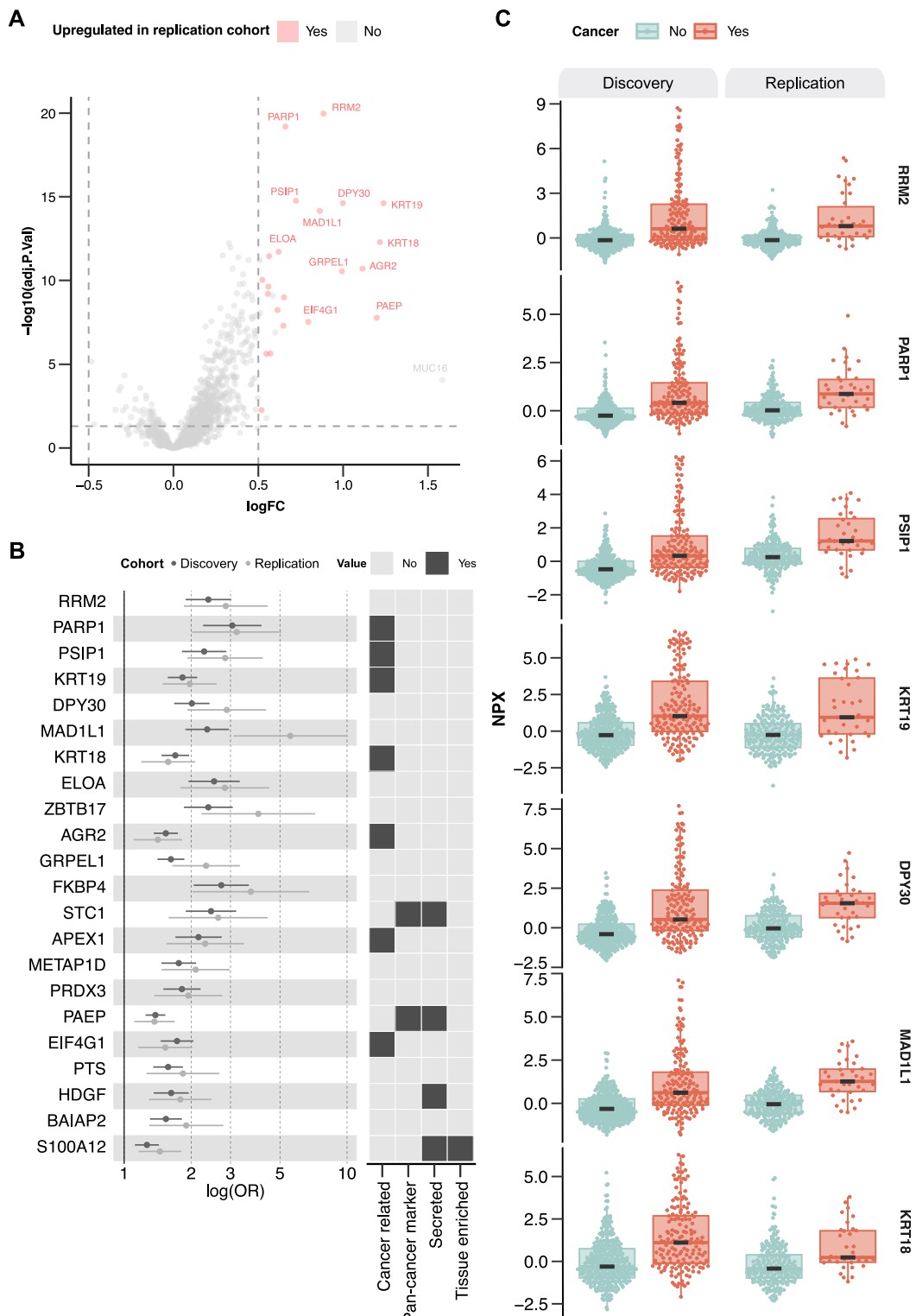

sample sizes and substantial heterogeneity within each group, these findings suggest that the model captures biologically relevant differences between cancer and other clinical conditions.

## Discussion

Screening for cancer using minimally invasive and easily accessible blood samples is an attractive approach with the advantage of high throughput, cost effective and objective assessments. However, due to inherent challenges such as tumor heterogeneity and study designs that do not accurately reflect real life settings, no such tests are currently available for pan-cancer screening. In this study, we developed a cancer prediction model for patients presenting with non-specific cancer symptoms using plasma levels of 1463 proteins quantified by proximity extension assay-based proteomics in pre-

**Fig. 2 | Differential expression analyses comparing patients later diagnosed with cancer to non-cancer in patients with non-specific symptoms of cancer. A** Volcano plot summarizing differential expression results comparing cancer and non-cancer patients in the discovery cohort ($n = 456$, cancer = 160, non-cancer = 296). Proteins upregulated also in the replication cohort are colored in red. *P*-values were derived from two-sided moderated *t*-tests from limma linear models and adjusted for multiple testing using the Benjamini–Hochberg method. **B** Left: Forest plot displaying odds ratios (ORs) per 1 NPX increase and 95% confidence intervals (log scale) from logistic regression models adjusted for age and sex, for proteins differentially expressed between cancer and non-cancer patients. Error bars represent the 95% confidence intervals, and data are centered on the estimated OR. Corresponding *p*-values are listed in Supplementary Table S5. Right:

Characterization of the differentially expressed proteins according to whether they are cancer-related, secreted or tissue enriched according to the Human Protein Atlas annotation (v24.proteinatlas.org), or associated with pan-cancer according to Álvez et al.[29]. **C** Protein levels of the top significant differentially expressed proteins in the discovery cohort, presented in the discovery ($n = 456$, cancer = 160, non-cancer = 296) and replication cohorts ($n = 238$, cancer = 35, non-cancer = 203). Box plots display the median (center line), interquartile range (box bounds), and whiskers representing the 1.5× interquartile range. Statistical analyses were performed on data derived from independent biological samples. No technical replicates were included. Sample sizes (*n*) indicate the number of individuals included in each analysis.

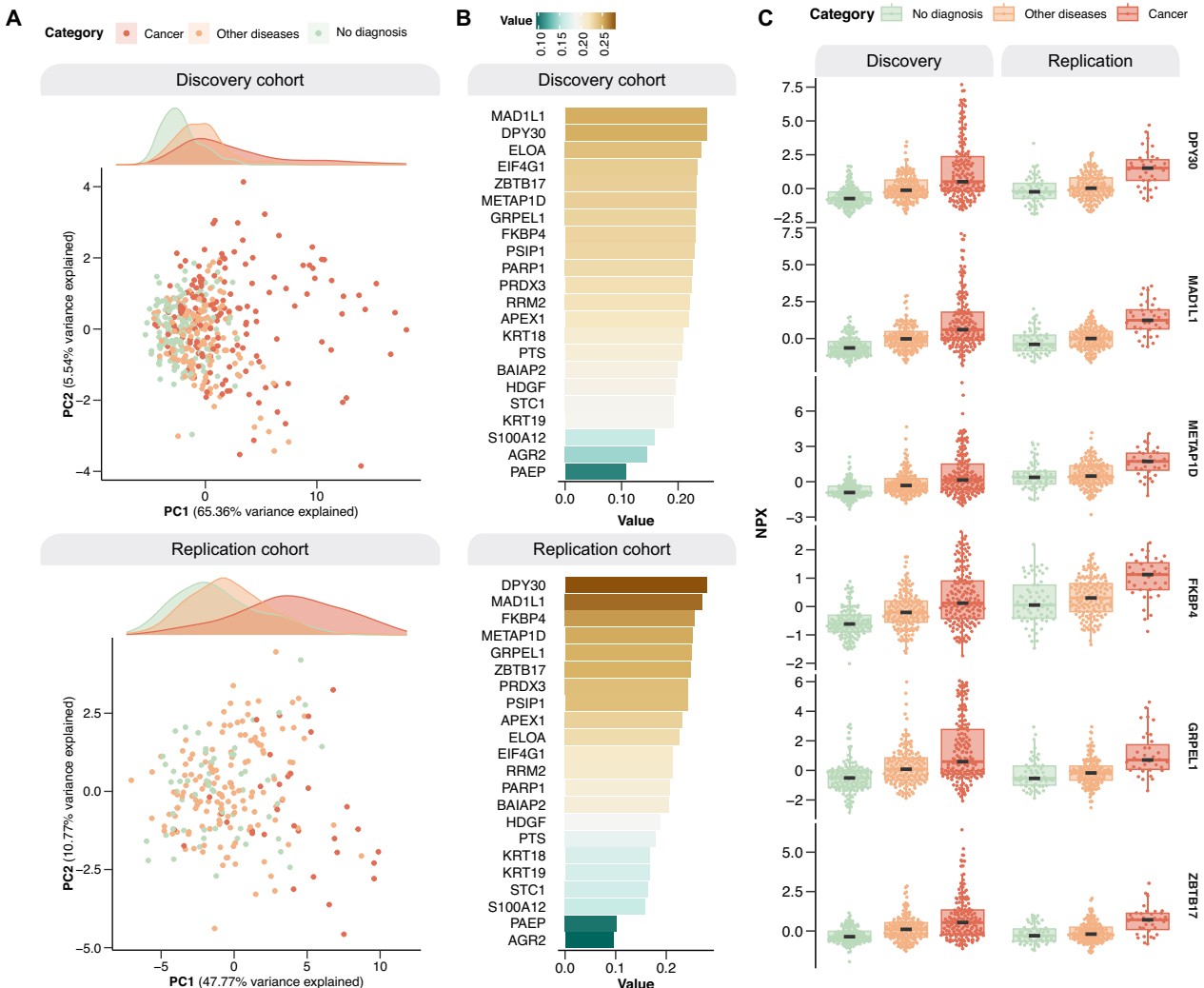

**Fig. 3 | PCA based on the differentially expressed proteins ($n = 22$) in the discovery and replication cohorts. A** PCA visualizations in the discovery and replication cohorts. **B** Loading scores of individual proteins contributing to PC1. **C** Boxplots showing protein expression across patient groups of selected proteins in the discovery cohort (no diagnosis = 141, other diseases = 155, cancer = 160) and replication cohort (no diagnosis = 83, other diseases = 120, cancer = 35). Patients with 'no

diagnosis' were either given a functional or non-somatic diagnosis, were diagnosed with a benign tumor, or had unexplained symptoms after evaluation. Box plots display the median (center line), interquartile range (box bounds), and whiskers representing the 1.5× interquartile range. All statistical analyses were performed on data derived from independent biological samples. No technical replicates were included. Sample sizes (*n*) indicate the number of individuals included in each analysis.

diagnostic samples. This model not only distinguished between patients later diagnosed with cancer and non-cancer symptomatic controls, but also differentiated patients later diagnosed with cancer from non-cancer symptomatic patients later diagnosed with auto-immune, inflammatory and infectious diseases. Remarkably, the

model maintained its performance in an independent replication cohort prior to diagnostic work-up. Although this classification model does not meet the demands of clinical implementation yet, it surpasses previous pan-cancer screening attempts and is a promising lead for further investigations.

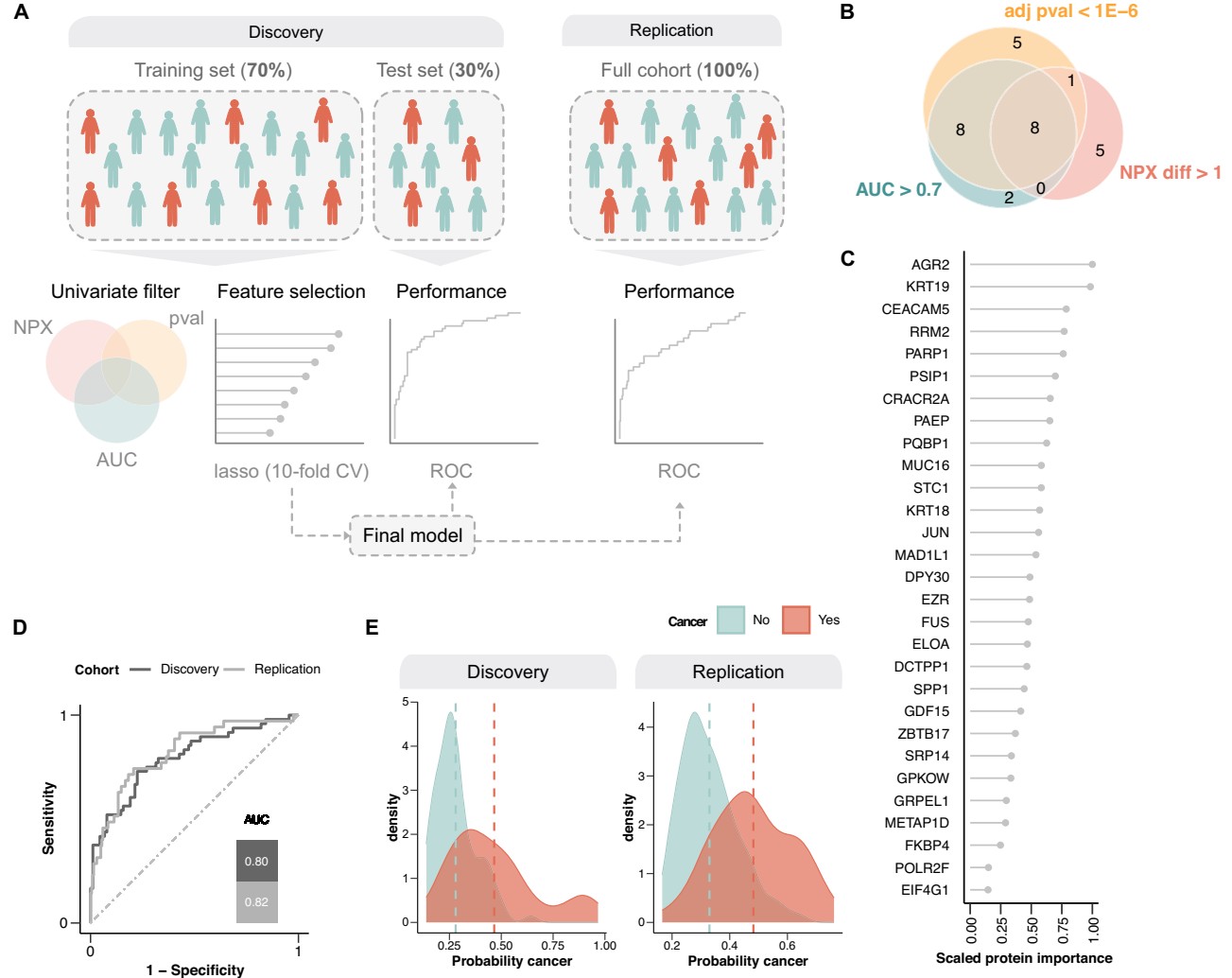

**Fig. 4 | Performance of the classification model discriminating cancer from non-cancer in patients presenting with non-specific symptoms of cancer.** **A** Overview of the classification model pipeline. **B** Venn-diagram showing the results of univariate feature selection in the training set within the discovery cohort. *P*-values were obtained using two-sided *t*-test and adjusted for multiple testing using the Benjamini–Hochberg method. **C** Key features selected by the lasso model. **D** ROC curves in the discovery and replication cohorts. **E** Distribution of model probabilities for cases and controls in the discovery and replication cohorts. The dashed lines represent the median cancer probability.

The majority of circulating cancer biomarker candidates never enter clinical practice[28,29]. Developing cancer biomarkers distinguishing cancer patients from patients with non-malignant inflammation is an even greater challenge, and one of the major strengths of this study is the use of two independent cohorts comprising symptomatic controls with inflammatory and autoimmune diseases. The control group in the current study presented with similar symptoms as patients diagnosed with cancer during the follow-up time and these patients were diagnosed with other non-malignant diseases, such as autoimmune, inflammatory and infectious diagnoses which are disease entities that can be difficult to distinguish from cancer. Another key strength of this study is the prospective cohort design with samples collected before cancer diagnostic work-up, cancer diagnosis, and treatment. Accordingly, while previously studied biomarkers[9] displayed higher AUC in predicting (pan-)cancer as compared to the current study, this may be a consequence of study design and control group composition.

Our model furthermore demonstrated a robust ability to differentiate between patients later diagnosed with cancer and patients later diagnosed with the various non-malignant disease entities, including autoimmune, inflammatory, infectious diseases and symptomatic patients with no diagnosis. As expected, AUC was highest for cancer vs no diagnosis (0.86), followed by an AUC value of 0.76 for cancer vs infectious disease, 0.75 for cancer vs autoimmune disease and 0.67 for cancer vs inflammatory disease. These results highlight the model's strong discriminatory power, particularly in distinguishing patients later diagnosed with cancer from patients later diagnosed with non-malignant infectious and autoimmune diseases. The slightly lower AUC for discriminating patients later diagnosed with cancer and patients later diagnosed with inflammatory diseases suggests a need for further refinement in this area and supports the evidence of crosstalk between cancer and inflammation, highlighting the challenges in distinguishing between these two conditions[27].

Several of the proteins identified by the machine learning model have been implicated in various tumor types. The six most important proteins according to the lasso model include Anterior gradient 2 (AGR2), Cytokeratin 19 (KRT-19), Carcinoembryonic antigen-related cell adhesion molecule 5 (CEACAM5), Ribonucleotide reductase subunit M2 (RRM2), Poly ADP-ribose polymerase 1 (PARP1) and PC4 and SFRS1 interacting protein 1 (PSIP1). AGR2, which plays a role in several cellular processes such as migration, differentiation and proliferation,

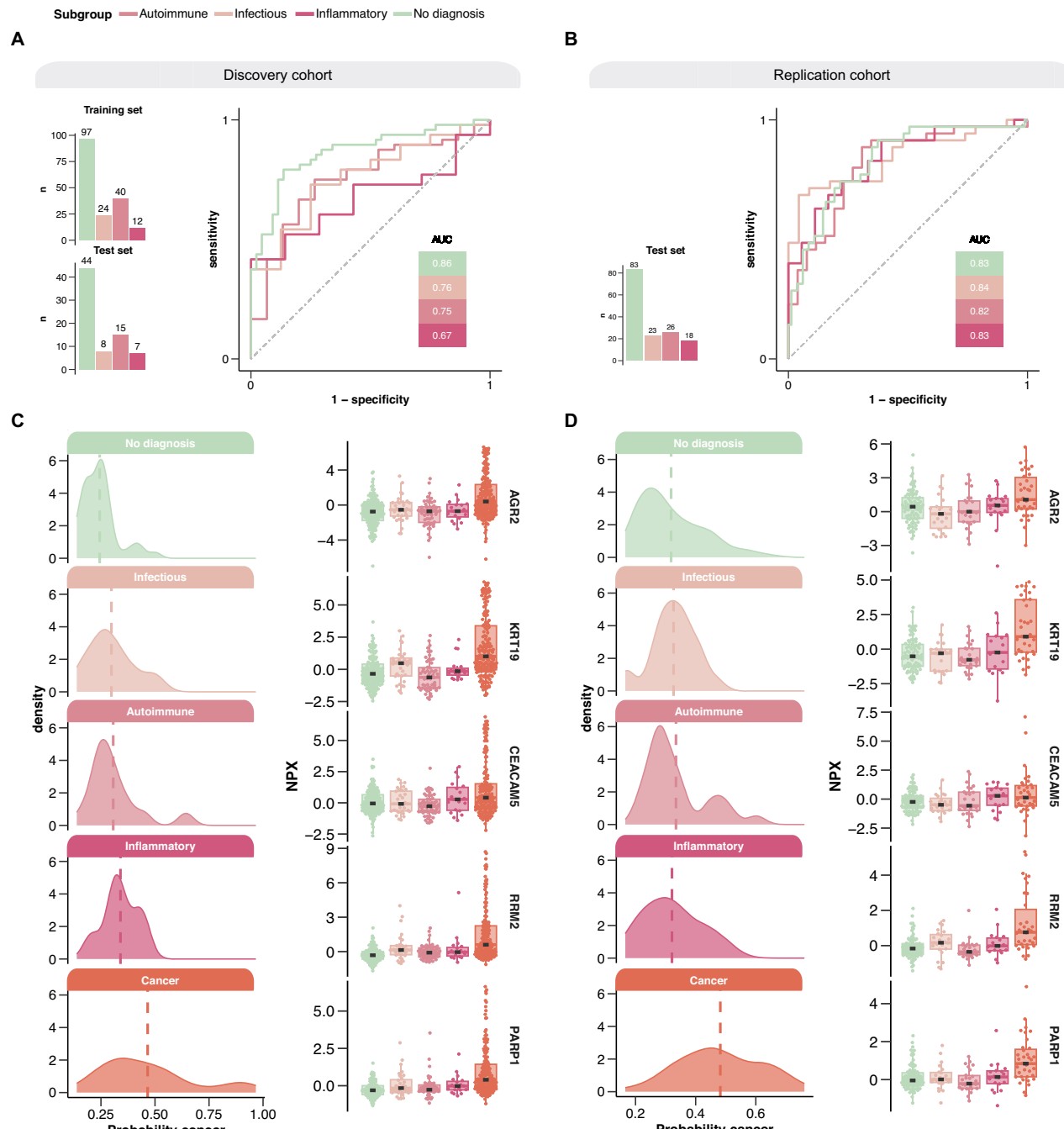

**Fig. 5 | Performance of the pan-cancer classification model to discriminate between cancer and different disease entities in patients presenting with non-specific symptoms of cancer.** The model's ability to differentiate patients later diagnosed with cancer from patients with no diagnosis, infectious, autoimmune, and inflammatory disease is illustrated using ROC curves in the discovery (no diagnosis = 141, infectious = 32, autoimmune = 55, and inflammatory disease = 19) (**A**) and replication cohort (no diagnosis = 83, infectious = 23, autoimmune = 26, and inflammatory disease = 18) (**B**). The distribution of model probabilities and the five most important proteins in the model for discovery cohort (no diagnosis = 141, infectious = 32, autoimmune = 55, inflammatory disease = 19, and cancer n = 160 (**C**) and replication cohort (no diagnosis = 83, infectious = 23, autoimmune = 26, inflammatory disease = 18, and cancer n = 35 (**D**). The dashed lines represent the median cancer probability. Patients categorized as "no diagnosis" include those with functional or non-somatic diagnoses, benign tumor conditions that did not explain their symptoms or cases with unexplained symptoms/pathological findings after the diagnostic work-up. Box plots display the median (center line), interquartile range (box bounds), and whiskers representing the 1.5× interquartile range. All statistical analyses were performed on data derived from independent biological samples. No technical replicates were included. Sample sizes (n) indicate the number of individuals included in each analysis.

has been identified as a pro-oncogenic protein that inhibits p53-activity. It has been observed in several solid tumors such as pancreatic, breast, lung, ovarian, prostate and colorectal cancer[30]. KRT-19 is expressed in epithelial tissues and is predominantly found in

proliferating regions in the gastrointestinal tract and has been shown to be overexpressed in radio-resistant colon tumors[31]. CEACAM5, also known as CEA, is a member of the CEA-family and is localized in gastrointestinal tract, cervix, sweat glands and prostate[32]. It plays

important roles in cell adhesion, intra- and intercellular signaling as well as cancer progression, inflammation, angiogenesis and metastasis. CEACAM5 is used for disease monitoring in colorectal, lung, breast, gastric, pancreatic and liver cancer but has also been associated with gallbladder, urinary bladder, mucinous ovarian and endometrium carcinomas[32]. RRM2 is an enzyme involved in DNA synthesis and repair and has been associated with poor prognosis in patients with lung adenocarcinoma[33]. PARP1, a nuclear enzyme involved in DNA repair and gene transcription, is present in a variety of tumor tissues and is targeted by inhibitors used for treatment of several cancer types such as ovarian, breast, pancreatic and prostate cancers[34]. PSIP1, a transcriptional coactivator, has been associated with increased tumorgenicity in breast cancer[35].

Notably, all proteins were found to be consistently upregulated using the selected cutoffs in the differential expression analyses in both cohorts. One reason for this could be that the plasma proteomics assay used in this study primarily captures changes in the circulatory blood proteome, which limits its ability to detect the down- or upregulation of other proteins, such as intracellular compartments or membrane-associated proteins often hallmarking cancer[36–41].

The most common cancer type identified in our cohort was hematologic malignancies, which reflects the nature of the study population that includes patients with non-specific symptoms who are not eligible for organ-specific diagnostic pathways. Patients presenting with organ-specific symptoms, such as a breast lump or rectal bleeding, or radiological signs suggestive of organ-specific cancers are typically referred directly to corresponding pathways and were therefore not included in this study. As a result, our cohort, recruited from a fast-track diagnostic out-patient clinic for unspecific cancer symptoms, includes a higher proportion of diagnostically challenging cancers, such as hematologic malignancies, for which the initial clinical presentation lacks clear organ-specific indicators. This explains the overrepresentation of certain cancer types compared to the general population and underscores the clinical complexity of the target group addressed by our model.

There are several limitations worth addressing. Firstly, this study is limited by the large proportion of patients later diagnosed with metastatic disease, hampering generalizability in cohorts with a larger proportion of localized disease. Secondly, the replication cohort has a lower proportion of patients later diagnosed with cancer because it does not include patients with radiological findings of metastases without an identifiable primary tumor. However, the model's ability to perform with equal performance in the replication cohort despite this difference is a great advantage. Thirdly, the sensitivity of the proteins to pre-analytical variation should also be considered for further investigations. Finally, direct comparison with commercial pan-cancer tests, such as Galleri (GRAIL)[11] and Cancerguard (formerly CancerSEEK)[8], was not performed and could therefore not be used as a benchmark. However, our model was specifically designed for triage in symptomatic individuals, which differs from the early screening focus of currently available commercial pan-cancer tests evaluated in asymptomatic populations.

Establishing blood-based diagnostic biomarkers that exceed or match the effectiveness of conventional cancer diagnostics, such as radiology and biopsy, poses a great challenge due to the high level of accuracy these reference methods provide. While the current model does not rival the precision of established advanced imaging such as positron emission tomography-computed tomography scan, it demonstrates promising performance (AUC 0.80–0.82 in both discovery and replication cohorts) and is designed to support earlier clinical decision-making, particularly in primary care. Patients presenting with vague or non-specific symptoms often fall outside organ-specific cancer pathways, and referring all such individuals for advanced imaging would place unsustainable demands on healthcare resources while exposing patients to unnecessary radiation and invasive procedures. By incorporating proteomic profiles into a triage framework, our model could help identifying patients at higher risk of malignancy and guide timely diagnostic work-up, including imaging and biopsy, where appropriate.

This triage approach is particularly valuable for general practitioners, who frequently face the challenge of differentiating between malignant and benign causes of symptoms such as fatigue, weight loss, or pain. The model's "rule-in" utility is supported by the predicted probability distributions shown in Fig. 5, where cancer patients cluster toward the higher end of the probability scale in both cohorts. This suggests that the model is well suited for identifying patients likely to benefit from a rapid evaluation, especially those who may not initially qualify for specialized diagnostic pathways. While the current study was conducted in a high-prevalence cohort, future validation in lower-prevalence primary care populations is essential to assess generalizability. Overall, leveraging validated protein biomarker profiles into actionable proteomic risk scores could aid in streamlining referral decisions and reducing missed cancers. As research advances, we believe that the clinical application of proteomic risk scores is poised to revolutionize cancer management, offering a powerful tool for clinicians in the fight against cancer by identifying patients who could benefit from a rapid diagnostic work-up using orthogonal methods.

Looking ahead, several steps are needed to support clinical translation and broader implementation of this approach. Firstly, validation in primary care populations with lower cancer prevalence will be crucial to assess generalizability and clinical utility in real-world decision-making. Secondly, direct comparison with other diagnostic blood test based platforms, including commercial pan-cancer tests such as GRAIL's Galleri[11] and Cancerguard (formerly CancerSEEK)[8] will be important to contextualize performance and added value. Additionally, the development of a simplified protein panel based on a smaller number of high-performing biomarkers, quantified using absolute protein concentrations to facilitate clinical implementation, would be of great value. Finally, validation of the model in healthy population cohorts will help to evaluate the performance in asymptomatic individuals. Together, these future efforts will provide critical insights for refining the model and advancing its clinical applicability in early cancer detection and diagnostic triage.

In summary, our study identifies several plasma proteins that can distinguish cancer from other inflammatory conditions using samples collected from patients before cancer diagnostic work-up and cancer diagnosis. Furthermore, we developed a model which discriminates cancer from infectious, autoimmune, and inflammatory diseases and confirm its capacity in an independent cohort also presenting with unspecific symptoms. Although further studies are needed in clinical settings with a lower prevalence of cancer, this study emphasizes the potential of blood proteome profiling to transform clinical medicine.

## Methods

Both MEDECA (discovery cohort) and ALLVOS (replication cohort) studies comply with the Declaration of Helsinki and all patients provided written informed consent. The MEDECA study was approved by the regional ethical review board in Stockholm and the Swedish Ethical Review Authority (dnr 2017/2160-31/1, 2019-00677, 2020-00186, 2021-02939 and 2023-07877-02) while the ALLVOS study was approved by the regional ethical review board in Uppsala and the Swedish Ethical Review Authority (dnr 2018/082, 2022-05947-02, 2023-07972-02).

### Study population

**The MEDECA study.** A total of 872 patients were admitted to the fast-track multidisciplinary diagnostic pathway at the Diagnostic center at Danderyd Hospital, Stockholm Sweden, between March 2018 and September 2020. During this timeframe, 502 patients were enrolled in

the study, with temporary pauses during vacation periods and in the initial phase of the covid-19 pandemic. Forty-six patients were excluded from further analyses, either due to confirmed active cancer prior to inclusion ($n = 21$), lost to follow-up ($n = 4$), withdrawal of consent ($n = 2$), insufficient plasma samples ($n = 18$) or identified as outliers ($n = 1$, outlier in majority of measured proteins, likely due to large monoclonal IgG paraprotein). No statistical method was used to predetermine sample size. Inclusion criteria were either Non-specific Signs and Symptoms of Cancer (NSSC) or radiological sign of malignancy without an apparent primary site and therefore not eligible for entrance into an organ-specific pathway. NSSC onset was required to be 6 months or less prior to referral and without any other obvious explanation. They were defined as any of the following symptoms: general malaise, extreme fatigue, reduced appetite, unintentional weight loss of more than 5 kg, prolonged fever, unexplained pain, pathological laboratory values (such as anemia, elevated alkaline phosphatase, erythrocyte sedimentation rate or calcium levels), increased contacts to health system or increased use of medications. Patients presenting with organ-specific symptoms are typically referred directly to corresponding fast-track diagnostic pathways and, as such, are admitted to the diagnostic out-patient clinic and therefore not included in this study. All study patients underwent a standardized and extensive cancer diagnostic work up (i.e., fast-track pathway), including an expanded panel of biochemical analyses, diagnostic tissue biopsies and imaging such as computed tomography, magnetic resonance or ${}^{18}$F fluorodeoxyglucose positron emission tomography/computed tomography investigations. Demographic data, comorbidity, cancer diagnosis and other diagnoses were obtained from hospital records.

**The ALLVOS study.** A total of 280 patients admitted to the fast-track pathway for patients with unspecific cancer symptoms at Örebro University Hospital between October 2018 and December 2022 were asked to participate in the study. During this timeframe, 248 patients accepted and were enrolled in the study. Inclusion criteria were identical with the MEDECA study but did not include patients with radiological sign of malignancy without an apparent primary site. Patients with confirmed active cancer at inclusion ($n = 2$), insufficient plasma samples ($n = 7$) and lost to follow up ($n = 1$) were excluded from further analyses. No statistical method was used to predetermine sample size. Clinical data was obtained from hospital records.

### Outcome
Patients were followed for six months from study inclusion regarding new cancer diagnosis. Diagnoses were obtained from individual medical records. Basal cell carcinoma diagnosis was not classified as cancer as it does not share the metastatic features of other cancers.

### Measurement of protein levels
Venous blood samples were collected at enrolment at the first visit to the Diagnostic center (i.e. before cancer diagnostic work-up). For MEDECA, EDTA plasma samples were centrifuged for 20 min at $2000 \times g$ at room temperature immediately following sampling and were stored at −80 °C until further analysis. For ALLVOS, EDTA plasma samples were centrifuged for 7 min at $2400 \times g$ at room temperature and frozen at −80 °C until further analysis within 4 h of sampling. Protein levels were analyzed using the proximity extension assay (PEA) by Olink at SciLifeLab Affinity Proteomics Uppsala (54). In PEA, matched pairs of oligonucleotide-labeled antibodies will bind to their target antigens in a pairwise manner. Upon antibody binding, the matched oligonucleotides are brought into proximity and with the use of a DNA polymerase, a PCR target sequence is created, amplified, detected, and quantified using NGS. The Olink Explore 1536 panel used in this study is comprised of four sub panels: Olink Explore 384 Cardiometabolic (LOT number B04413), the Explore 384 Inflammation (LOT number B04411), Olink Explore 384 Oncology (LOT number B04412) and Explore 384 Neurology (LOT number B04414). In total, 3.7 µl plasma per sample was used.

Following sequencing on an Illumina NovaSeq 6000, the Olink Explore platform generated raw count data where each assay/sample pair was assigned an integer value representing the number of detected DNA barcode copies. These raw counts were converted to Normalized Protein eXpression (NPX) values, which are relative quantification values on a log2 scale and used for downstream analyses. The NPX calculation involved two main steps. First, the assay-specific counts for each sample and assay block were normalized to the corresponding Extension Control signal and subsequently log2-transformed. The Extension Control consists of a matched antibody pair tagged with complementary DNA oligonucleotides that are always in proximity, independent of antigen binding. It provides a stable reference for controlling variation in the extension and amplification steps. Additional internal controls, including the Incubation Control and Amplification Control, were used to monitor other aspects of the assay workflow but were evaluated separately as part of quality control (QC) and not included in the actual NPX calculation.

Because samples were fully randomized across plates, it was assumed that majority of proteins were not differentially expressed between plates. Therefore, Intensity Normalization was applied as a between-plate normalization method. In this approach the median NPX per assay across all QC-passed, non-control samples on each plate is centered to zero. This zero-centered value then serves as the reference for aligning NPX distributions across plates, effectively removing systematic intensity differences and enabling comparability across the dataset. Higher NPX equals relatively higher protein abundance, negative NPX equals lower than the reference level and a difference of one NPX means a doubling for the protein concentration. All data processing and normalization steps were performed using Olink's proprietary software package NPX Explore HT and 3072.

Additional quality control of the dataset included removing samples where more than 50% of the proteins failed quality control, excluding individual protein measurements that failed quality control, and retaining one assay for proteins measured across all four panels (TNF, IL-6, and CXCL8).

### Statistical analyses
Statistical analyses were performed in R version 4.2.1[42]. Differential expression analyses were performed using the limma package (version 3.54.0)[43], including age and sex as covariates, using a log fold change (logFC) cut-off of 0.5. Resulting *p*-values were corrected for multiple hypothesis testing using the Benjamini–Hochberg method[44] and a *p*-value of 0.05 was used as threshold for significance. The tidymodels (version 1.0.0)[45] package was used for PCA visualization and for model development. Logistic regression models adjusted for age and sex were used to estimate the association between individual protein levels and cancer risk, implemented using the glm engine. Multivariate regression models with lasso regularization were used to identify predictive protein signatures, implemented via the glmnet engine. For these analyses, the input data from the discovery cohort was split into a training and test set in a 70/30 ratio. Within the training set, we applied three initial filtering criteria to reduce the numbers of features prior to model fitting, where a protein needed to meet at least one: (i) a mean NPX difference greater than 1, (ii) an adjusted *p*-value < 1E−6 from a *t*-test, or (iii) an area under the receiver operating curve (AUC) > 0.7 for individual proteins. The model-based feature selection and hyperparameter optimization were performed using the training set data within a 10-fold cross

validation scheme. The performance of the generated model was estimated in the test set as well as an independent replication cohort using the area under the ROC curve (AUC) metrics. Protein importance is defined by the protein estimates in the lasso model, which were scaled from 0 to 1 and are referred to as "scaled protein importance." Data visualization was performed using ggplot2 (version 3.4.0)[46], ggbeeswarm (version 0.7.1)[47], ggforestplot (version 0.1.0)[48], ggrain (version 0.0.3)[49] ggrepel (version 0.9.2)[50] ggridges (version 0.5.4)[51] patchwork (version 1.1.2)[52], and pheatmap (version 1.0.12)[53] R packages. The figures were assembled in Affinity designer 2 (version 2.5.0).

## Statistics and reproducibility

All statistical analyses were performed in R version 4.2.1[42]. Associations between plasma protein levels and cancer diagnosis were assessed using logistic regression models adjusted for age and sex, and multivariate regression models with lasso regularization were used to identify predictive protein signatures. Model performance was evaluated using ROC curves and AUC values.

The discovery cohort included 456 prospectively enrolled patients, with validation in an independent similar cohort of 238 patients. No statistical method was used to predetermine sample size. Patients were excluded due to pre-existing cancer ($n = 23$), lost to follow-up ($n = 5$), withdrawal of consent ($n = 2$), insufficient plasma ($n = 25$), or outlier ($n = 1$). Samples were randomized across plates, and analyses were performed on independent biological samples without technical replicates. Results were reproducible across both cohorts.

## Reporting summary

Further information on research design is available in the Nature Portfolio Reporting Summary linked to this article.

## Data availability

The data set generated in this study contains pseudonymized (coded) personal data derived from human participants. A key code allowing re-identification is held by the health care provider. According to the EU General Data Protection Regulation (GDPR) and the conditions of the Swedish Ethical Review Authority approval, these data cannot be publicly shared. Access to the data is restricted to ensure compliance with GDPR and Swedish ethical and legal requirements. Data may be made available only for non-commercial research purposes to researchers, and only under a data access agreement that ensures protection of personal data and compliance with institutional and ethical guidelines. Requests for access can be sent to the corresponding author or Data Protection Officer at Danderyd Hospital (dso.ds@regionstockholm.se), who will assess eligibility, manage the legal/ethical review, and coordinate secure data transfer if approved. Requests will be reviewed as promptly as possible. The data will remain available for at least 10 years post publication to comply with the Swedish Archive Law.

## Code availability

All generated code is available at https://github.com/buenoalvezm/MEDECA[54].

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

## Acknowledgements

We thank Martha Kihlgren, Maja Månsson, Henry Ng and Lena Gabrielsson for their assistance with the laboratory work. We also acknowledge all clinicians, nurses and patients at the Diagnostic Centers for their valuable contributions to this study. This work was supported by grants from the Swedish Society for Medical Research (SSMF) Grant PG 23-0399 (A.R.), Research Residency Stockholm County Council (F.W.), Nyckelfonden Örebro University Hospital Research Foundation OLL-935156 (G.H.), the Regional Agreement on Medical Training and Clinical Research (A.L.F.) between Örebro County Council and Örebro University (A.Q. and E.A.), WCPR grant KAW2022.0318 from Knut and Alice Wallenberg Foundation (M.U.), the Jochnick Foundation (C.T.), The Swedish Research Council 2022-02699 (C.T.), the Regional Agreement on Medical Training and Clinical Research (A.L.F.) between Stockholm County Council and Karolinska Institutet FoUI-976394 (C.T.), and the Swedish Society of Medicine (C.T.).

## Author contributions

C.T., A.R. and V.H. initiated the MEDECA study and were responsible for study design. V.H. was responsible for patient inclusion. G.H., A.Q. and E.A. initiated the ALLVOS- study. F.W. and L.E. conducted the electronic medical chart review and developed the clinical database for the discovery cohort, while A.Q. and E.A. performed these tasks for the replication cohort. K.A. prepared the preanalytical samples with assistance from F.W. for the discovery cohort, while A.Q. and E.A. prepared the samples from the replication cohort. M.Å. oversaw the laboratory work. M.B.A. and M.G. performed the statistical analysis with assistance from F.W., A.R., M.Å., C.T. and T.P. M.B.A., F.W., C.T., A.R., T.P., V.H., M.Å., M.G., G.H., A.Q., E.A. and M.U. interpreted the data. F.W., M.B.A. and A.Q. drafted the first version of the manuscript. C.T., A.R., V.H., M.G. and G.H. supervised the study. All authors took part in reviewing and editing the manuscript and approved the final version of the manuscript.

## Funding

## Competing interests

The authors declare no competing interests.
