## [Transparent Peer Review file · Nature Communications]

Plasma protein profiling predicts cancer in patients with non-specific symptoms

Corresponding Author: Professor Charlotte Thålin

Version 0:

Reviewer comments:

Reviewer #1

(Remarks to the Author)

This is an intriguing study where the authors search for risk discriminatory protein biomarkers in a cohort of individuals scheduled for diagnostic work up of cancer. 1/3 of the patients were diagnosed with cancer during that work-up. The authors identified a number of markers apparently associated with risk, several of which were replicated in an independent similar cohort. The authors then developed a risk discrimination algorithm that performed well in both the training and replication dataset with AUCs around 0.8. The manuscript is well written and overall a nice read, although the intro should be tightened. The authors correctly highlight the importance of developing biomarkers in the target population, and I was pleased to see that the study managed to assemble quite similar and independent cohorts for training and testing which is rare. Whereas this is an important strength of the study, it is also a highly specific type of target population – individuals suspected to have cancer i.e. an extremely high risk population.

This leads me to one of the key issues of the study – what is the benchmark performance of biomarkers in this population? Whereas 0.8 in AUC would seem impressive, how does that compare with e.g. PET-CT? As currently fashioned, the implications of the study are unclear. It would seem pertinent to endeavour to combine, or at least compare, the protein algorithm with diagnostic procedures subsequent to the blood draw. Does it provide redundant, independent or complementary information to the diagnostic tests currently being used – as the paper reads now I don't know. In other words, I don't know if the study is important. Please explain.

Specific comments that require addressing by the authors

- Please address the comment in the last paragraph in the overall comments above. This is key.
- Intro can be tightened
- In the results, the authors present log fold-change between cases and non-cases as a measure of association for the individual markers. This is often done in studies comparing biomarkers between groups, however, as a measure of association of risk it is unclear as it only indicates the biomarker difference between the two groups. The research question here is (or should be), how much higher risk do people with a certain increase in biomarker-level have after taking obvious demographic factors into account (e.g. age and sex). This is expressed as relative risks associated with an e.g. doubling in biomarker concentrations and can be estimated by calculating hazard ratios (using adjusted cox-regression) or odds ratios (using logistic regression), with appropriate adjustments (e.g. age and sex). It seems that the study cohort is essentially a random cohort of really high-risk individuals, and thus time-to-event risk analysis should be appropriate (e.g. through Cox regression).
- Figure 2b can be replaced with a table with OR/HR as appropriate (see above), along with 95% CI.
- The construction of the risk model is not sufficiently detailed in the main manuscript – GLMNET is a package in a statistical software, not the statistical method in its own right. Please explain what type of model was used, linear, logistic, time-to-event (this would seem appropriate for the application), and what parameters were included in the final model (e.g. age and sex would be the minimal) in the main manuscript, not only in the online supplement.
- The cancer-specific signals tying back to the authors' previous work is not really warranted here. Considering the sample

size there is limited possibility to start to distinguish the overall cancer signal to cancer-specific signals. I suggest removing this analysis.

- Related to the the previous point, I think the main clinical concern addressed for this target population is to rule out cancer, ideally in combination with or by replacing (theoretically) other diagnostic tests. The sample size and methods would never be enough to do anything more specific, although this would be useful by itself. However, I'm not entirely sure of how generalizable this population is for this purpose. Please comment, and if the authors agree, consider introducing the ruling out as a potential use of this algorithm.
- The study represents is a very high risk prospectively followed study population which allows for time-to-event modelling, including calculation of absolute risks. (i.e. construction of an actual risk model rather than a classification algorithm).
- What are the next steps? Please explain more comprehensively what studies would be needed to translate this to clinical practice, the primary purpose (e.g. ruling out cancer perhaps), as complement/replacement to existing tools, establishing of custom panel with absolute protein concentrations, absolute risk modelling, study sample needed etc.

(Remarks on code availability)

Reviewer #2

(Remarks to the Author)

Using a proteomics approach by evaluating protein levels in blood plasma, the authors attempt to classify and stratify patients by cancer diagnosis and other possible malignant factors. The cohort consisted of individuals in the fast-track multidisciplinary cancer diagnostic pathway in Danderyd Hospital, Sweden. It was 502 in the discovery and 248 in the replication cohort. This seems interesting since it could be a diagnostic approach to cancer that would lead to less invasive and unnecessary procedures.

Strengths:

- 1) The paper has many strengths including a clear target population (those with cancer symptoms) and a clear application (rapidly segmenting patients into disease state).
- 2) The limitations of the test are clearly stated (except for point 2 below).

Weaknesses:

- 1) The distributions of detected cancers are somewhat surprising. Some explanation as to this distribution is necessary (e.g. everyone with breast cancer was identified earlier because of reasons 1 and everyone with colorectal cancer was detected because of reason 2).
- 2) One limitation would be to know how other commercial pan-cancer tests do on these same patients or at least a subset. I suspect some of these tests might do *very* well in this setting. I understand that these samples likely do not exist any longer but it would have made the study more valuable to understand other tests that are currently in the market.

(Remarks on code availability)

Reviewer #3

(Remarks to the Author)

Wannberg et al. present a well-designed study exploring the use of plasma proteomic profiling of over 1400 proteins for pan-cancer detection in patients presenting with non-specific symptoms. The study is novel and addresses an important clinical challenge in a realistic diagnostic context. However, potential clinical application of the test with the accuracy presented here is ultimately unclear, and additional development to reach higher accuracy would likely be needed.

The manuscript is well written and the figures are clear.

Major comments:

- On line 234, it is stated that "Prior to feature selection within the training set, we applied three criteria: ...". This wording is a bit ambiguous and could technically mean that the three filtering criteria were applied based on data from the entire discovery cohort "prior to feature selection within the training set". It would be good to clarify this point, so readers can feel more certain that the three criteria were only applied based on data from the training set.
- Relating to the previous point, saying that the three criteria were applied "prior to feature selection" feels slightly confusing, since application of the three criteria is very much a form of feature selection as well. Would be better to phrase them as two different types of feature selection.
- The Discussion states "Screening for cancer using minimally invasive and easily accessible blood samples is an attractive approach However, ..., no such tests are currently available for pan-cancer screening." But pan-cancer blood tests such as GRAIL Galleri are already available and (I believe) more accurate than the results presented here (for metastatic tumors).
- The text states that protein expression values are provided in normalized (NPX) units, but the text does not explain how those values are calculated (e.g. how are protein expression values normalized between samples?)
- Volume of blood used for analysis is not specified in Methods
- Instrument used for NGS is not specified in Methods

Minor comments:

- Typo on line 241: "contriubuted" -> "contributed"
- Typo on line 242: "proteins proteins"

(Remarks on code availability)

Version 1:

Reviewer comments:

Reviewer #1

(Remarks to the Author)

The authors have provided thoughtful answers to my comments. The only issue that needs fixing is to include the scale for the OR estimates wherever it's mentioned, including in the figures. - i.e. the odds ratio per (e.g.) standard deviation increment/doubling/whatever in the xxx concentrations.

(Remarks on code availability)

Reviewer #2

(Remarks to the Author)

The authors have not fairly represented the state of the MCED test space and its application to triage "We thank the reviewer for this suggestion. Although direct comparison with commercial pan-cancer tests ... such comparisons are currently not feasible due to limited access to those assays and limited sample material." At least in the US, and I suspect in Europe, Galleri for example is commercially available, all that would be necessary is to have a clinician order it and pay for it. I accept that the material might not be available but the test is, and has been for some time.

Further, although MCEds are designed to screen for early cancers, their actual training was on diagnosed cancers making it quite similar to the triage condition presented here. If an MCED can find an asymptomatic cancer, it could nearly certainly find a symptomatic cancer and it is my impression that MCEds are being used, at least occasionally, in this context.

(Remarks on code availability)

Reviewer #3

(Remarks to the Author)

The authors have satisfactorily addressed my comments.

(Remarks on code availability)

Reviewer #1 (Remarks to the Author):

This is an intriguing study where the authors search for risk discriminatory protein biomarkers in a cohort of individuals scheduled for diagnostic work up of cancer. 1/3 of the patients were diagnosed with cancer during that work-up. The authors identified a number of markers apparently associated with risk, several of which were replicated in an independent similar cohort. The authors then developed a risk discrimination algorithm that performed well in both the training and replication dataset with AUCs around 0.8. The manuscript is well written and overall a nice read, although the intro should be tightened. The authors correctly highlight the importance of developing biomarkers in the target population, and I was pleased to see that the study managed to assemble quite similar and independent cohorts for training and testing which is rare. Whereas this is an important strength of the study, it is also a highly specific type of target population – individuals suspected to have cancer i.e. an extremely high risk population.

This leads me to one of the key issues of the study – what is the benchmark performance of biomarkers in this population? Whereas 0.8 in AUC would seem impressive, how does that compare with e.g. PET-CT? As currently fashioned, the implications of the study are unclear. It would seem pertinent to endeavour to combine, or at least compare, the protein algorithm with diagnostic procedures subsequent to the blood draw. Does it provide redundant, independent or complementary information to the diagnostic tests currently being used – as the paper reads now I don't know. In other words, I don't know if the study is important. Please explain.

We appreciate the reviewer's thoughtful comment and the opportunity to clarify the clinical intention and potential utility of our proteomic model. As noted, the AUC of 0.8–0.82 observed in both discovery and replication cohorts indeed appears promising. However, we agree that the model is not meant to replace highly sensitive diagnostic modalities such as PET-CT. In fact, the reported AUC is evaluated against the reference cancer diagnosis established by the comprehensive diagnostic work-up including highly sensitive diagnostic modalities such as PET-CT. Instead, the purpose of our model is to support clinical decision-making earlier in the diagnostic pathway, particularly in a primary care setting. Importantly, patients presenting with vague and non-specific symptoms often fall outside organ-specific cancer pathways, and referring all such patients for advanced imaging like PET-CT would overload the healthcare systems burden patients with unnecessary radiation and invasive investigations.

Our classification model is designed to act as a triage tool. It can help identify those at higher risk of having an underlying malignancy and thus prioritize them for a rapid diagnostic work-up, including imaging and biopsies as appropriate. This approach could be particularly useful for GPs, who frequently face the challenge of distinguishing between benign and malignant causes of non-specific symptoms such as fatigue, weight loss, or unexplained pain. In this way, our model provides complementary information to existing diagnostic methods. Rather than functioning as a diagnostic tool per se, it serves to guide the allocation of diagnostic resources more effectively,

potentially reducing diagnostic delays and improving outcomes in this challenging patient population.

We acknowledge that the potential clinical application and intended use of the model needs clarification, and we have now clarified this in the revised abstract (line 48-52), introduction (line 183-190) and discussion (line 558-613).

Specific comments that require addressing by the authors

- Please address the comment in the last paragraph in the overall comments above.

This is key.

- Intro can be tightened

Thank you for this comment, please see answer above. We have shortened the introduction by removing detailed discussions of specific cancer biomarkers, relocating technical explanations of the method to the Methods section, and leaving out references to prior work on individual cancer types.

- In the results, the authors present log fold-change between cases and non-cases as measure of association for the individual markers. This is often done in studies comparing biomarkers between groups, however, as a measure of association of risk it is unclear as it only indicates the biomarker difference between the two groups. The research question here is (or should be), how much higher risk do people with a certain increase in biomarker-level have after taking obvious demographic factors into account (e.g. age and sex). This is expressed as relative risks associated with an e.g. doubling in biomarker concentrations and can be estimated by calculating hazard ratios (using adjusted cox-regression) or odds ratios (using logistic regression), with appropriate adjustments (e.g. age and sex). It seems that the study cohort is essentially a random cohort of really high-risk individuals, and thus time-to-even risk analysis should be appropriate (e.g. through Cox regression).

Thank you for this comment. We used student t-test to identify differentially expressed proteins. We agree that odds ratios (OR) would be more interpretable clinically. We have therefore reanalyzed the data using logistic regression adjusted for age and sex, to calculate ORs for cancer risk and replaced Figure 2b with ORs summarizing these associations with ORs, 95% CIs, and p-values. These results are now included in the main manuscript (Results section).

- Figure 2b can be replaced with a table with OR/HR as appropriate (see above), along with 95% CI.

Figure 2b has been updated to display ORs, and the corresponding exact values with 95% confidence intervals are now provided in the Supplementary Material (**Supplementary Table S5**).

- The construction of the risk model is not sufficiently detailed in the main manuscript – GLMNET is package in a statistical software, not the statistical method in its own right.

Please explain what type of model was used, linear, logistic, time-to-event (this would seem appropriate for the application), and what parameters were included in the final model (e.g. age and sex would be the minimal) in the main manuscript, not only in the online supplement.

We have now clarified in the methods and results that a multivariate logistic regression model with lasso regularization was used to predict cancer status based on protein levels.

- The cancer-specific signals tying back to the authors previous work is not really warranted here. Considering the sample size there is limited possibility to start to distinguish the overall cancer signal to cancer-specific signals. I suggest removing this analysis.

We acknowledge that the sample sizes for individual cancer types were limited and that robust cancer-type stratification was not the primary aim of this study. We have therefore removed the section on cancer-type specific analyses.

- Related to the the previous point, I think the main clinical concern addressed for this target population is to rule out cancer, ideally in combination with or by replacing (theoretically) other diagnostic tests. The sample size and methods would never be enough to do anything more specific, although this would be useful by itself. However, I'm not entirely sure of how generalizable this population is for this purpose. Please comment, and if the authors agree, consider introducing the ruling out as a potential use of this algorithm.

We agree that one of the most clinically meaningful applications of this model is its potential to rule in cancer among patients presenting with non-specific symptoms, particularly in primary care settings. Our model is designed to be applied to patients with low to moderate clinical suspicion, i.e those who may otherwise not qualify for a rapid diagnostic work-up. As only a subset of patients with suspected cancer are referred to specialized diagnostic outpatient clinics, this test could help streamline referral decisions and reduce the number of missed cancers by identifying individuals who would benefit from a rapid diagnostic work-up. The "rule-in" utility is supported by the probability distributions shown in Figure 5, where cancer patients cluster toward the upper end of the predicted probability range in both the discovery and replication cohorts. This indicates that the model assigns high probabilities to true cancer cases, suggesting it is well suited for flagging individuals at high risk of cancer. We have now clarified in the Discussion (line 569-613) that, while this study was conducted in a high-prevalence setting, future validation in lower-prevalence primary care populations will be critical to assess its utility in improving early cancer detection and optimizing diagnostic resource allocation.

- The study represents is a very high risk prospectively followed study population which allows for time-to-event modelling, including calculation of absolute risks. (i.e. construction of an actual risk model rather than a classification algorithm).

While we appreciate the suggestion to perform time-to-event analyses, we believe this approach is not appropriate in the context of our study design. The reported AUC is evaluated against the reference cancer diagnosis established by the comprehensive diagnostic work-up and all cancer patients therefore received a definitive cancer diagnosis within the diagnostic work-up period. In cases where the diagnosis took longer (e.g 4 months vs 1 month), the delay reflected diagnostic complexity, such as waiting time to biopsy and pathological examination. Therefore, time-to-diagnosis in our cohort is not an appropriate method to investigate cancer risk. Instead, we have focused on using logistic regression, which better reflects the clinical question of identifying high-risk individuals at the point of initial evaluation.

Kaplan Meier Curve showing proportion of patients diagnosed with cancer over 180 days from study inclusion.

- What are the next steps? Please explain more comprehensively what studies would be needed to translate this to clinical practice, the primary purpose (e.g. ruling out cancer perhaps), as complement/replacement to existing tools, establishing of custom panel with absolute protein concentrations, absolute risk modelling, study sample needed etc.

We now include a dedicated paragraph in the discussion outlining next steps, including:

- Validation in primary care settings with lower cancer prevalence
- Comparison with other diagnostic tests such as GRAIL
- Development of a simplified protein panel with absolute protein concentrations
- Validation of the model in existing healthy population cohorts

We also have ongoing discussions with Cancerguard (former CancerSEEK) to explore the use of our samples for validation of their test in a real-world diagnostic setting.

Reviewer #2 (Remarks to the Author):

Using a proteomics approach by evaluating protein levels in blood plasma, the authors attempt to classify and stratify patients by cancer diagnosis and other possible malignant factors. The cohort consisted of individuals in the fast-track multidisciplinary cancer diagnostic pathway in Danderyd Hospital, Sweden. It was 502 in the discovery and 248 in the replication cohort. This seems interesting since it could be a diagnostic approach to cancer that would lead to less invasive and unnecessary procedures.

Strengths:

- 1) The paper has many strengths including a clear target population (those with cancer symptoms) and a clear application (rapidly segmenting patients into disease state).
- 2) The limitations of the test are clearly stated (except for point 2 below).

Weaknesses:

- 1) The distributions of detected cancers are somewhat surprising. Some explanation as to this distribution is necessary (e.g. everyone with breast cancer was identified earlier because of reasons 1 and everyone with colorectal cancer was detected because of reason 2).

We appreciate the reviewer's positive assessment of our study's potential to support clinical decision-making while reducing unnecessary invasive diagnostics. However, we agree that the observed distribution of cancer types warrants clarification. Patients with organ-specific symptoms or organ-specific radiological findings are often referred directly to organ-specific diagnostic pathways (e.g., breast or colorectal cancer). These patients often present with symptoms such as a lump in the breast or blood in the stool and are therefore entitled to these cancer-specific pathways and not included in this study. As a result, this cohort from the diagnostic out-patient clinic represents patients without clear indications of organ-specific cancer, leading to an overrepresentation of cancers that are more diagnostically challenging, such as hematologic malignancies or neuroendocrine tumors. Rather than functioning as a diagnostic tool per se, it serves to guide the allocation of diagnostic resources more effectively, potentially reducing diagnostic delays and improving outcomes in this challenging patient population. We have expanded this explanation in the Methods (line 646-649) and Discussion (line 525-536) sections to clarify this point.

- 2) One limitation would be to know how other commercial pan-cancer tests do on these same patients or at least a subset. I suspect some of these tests might do *very* well in this setting. I understand that these samples likely do not exist any longer but it would have made the study more valuable to understand other tests that are currently in the market.

We thank the reviewer for this suggestion. Although direct comparison with commercial pan-cancer tests, such as Galleri by GRAIL and Cancerguard, would indeed strengthen the study, such comparisons are currently not feasible due to limited access to those assays and limited sample material. We have added a discussion on this limitation in the revised Discussion section (line 546-553) and outline the importance of future benchmarking studies using shared reference cohorts, particularly in real-world symptomatic populations such as ours. We also clarify that our model was designed

specifically for triage in symptomatic individuals, rather than early screening in asymptomatic individuals, which is the setting most commercial tests are evaluated in (line 551-553).

Reviewer #3 (Remarks to the Author):

Wannberg et al. present a well-designed study exploring the use of plasma proteomic profiling of over 1400 proteins for pan-cancer detection in patients presenting with non-specific symptoms. The study is novel and addresses an important clinical challenge in a realistic diagnostic context. However, potential clinical application of the test with the accuracy presented here is ultimately unclear, and additional development to reach higher accuracy would likely be needed.

We thank the reviewer for the encouraging words. The model is not intended to replace highly sensitive diagnostic tools like PET-CT, but rather to support earlier clinical decision-making, particularly in primary care. As only a subset of suspected cancer patients are referred to specialized outpatient diagnostic clinics, our model could help streamline referrals and reduce the number of missed cancers. In this way, our model provides complementary information to existing diagnostic methods. We agree that while the current model demonstrates promising performance, additional development is needed to enhance diagnostic accuracy and support clinical translation. We now discuss these limitations and outline next steps, including model refinement, clinical validation in broader populations, and integration with complementary diagnostic tools in the Discussion section (line 547-554, 559-614).

The manuscript is well written and the figures are clear.

Major comments:

- On line 234, it is stated that "Prior to feature selection within the training set, we applied three criteria: ...". This wording is a bit ambiguous and could technically mean that the three filtering criteria were applied based on data from the entire discovery cohort "prior to feature selection within the training set". It would be good to clarify this point, so readers can feel more certain that the three criteria were only applied based on data from the training set.

Thank you for pointing this out. To clarify, all three filtering criteria were exclusively applied within the training set, as part of a two-step feature selection process. We have revised the sentence in the methods:

"Within the training set, we applied three initial filtering criteria to reduce the numbers of features prior to model fitting, where a protein needed to meet at least one: (i) a mean NPX difference greater than 1, (ii) an adjusted p-value $< 1E-6$ from a t-test, or (iii) an area under the receiver operating curve (AUC) > 0.7 for individual proteins"

- Relating to the previous point, saying that the three criteria were applied "prior to feature selection" feels slightly confusing, since application of the three criteria is very much a

form of feature selection as well. Would be better to phrase them as two different types of feature selection.

We agree and have adjusted terminology to better distinguish filter-based preselection from the model-based feature selection performed via lasso regularization.

- The Discussion states "Screening for cancer using minimally invasive and easily accessible blood samples is an attractive approach However, ..., no such tests are currently available for pan-cancer screening." But pan-cancer blood tests such as GRAIL Galleri are already available and (I believe) more accurate than the results presented here (for metastatic tumors).

We appreciate the correction. Although commercial pan-cancer blood tests such as GRAIL Galleri are now available, however they are currently not approved and typically evaluated in asymptomatic screening populations and target a different clinical setting than ours. Our study instead focuses on patients with non-specific symptoms, aiming to stratify symptomatic individuals for appropriate diagnostic work-up. A strength of our study is the use of a symptomatic control group, including patients later diagnosed with autoimmune, inflammatory and infectious conditions, which reflects a real-world clinical setting. This is now explained in the Discussion (line 547-554, 559-581).

- The text states that protein expression values are provided in normalized (NPX) units, but the text does not explain how those values are calculated (e.g. how are protein expression values normalized between samples?)
- Volume of blood used for analysis is not specified in Methods
- Instrument used for NGS is not specified in Methods

We apologize for not being clear on this. We have now re-written this section in Methods (line 683-731) to provide more details concerning the calculation and normalization of the NPX data, volume of blood used for analysis and the instrument used for NGS:

“Venous blood samples were collected at enrolment at the first visit to the Diagnostic center (i.e. before cancer diagnostic work-up). For MEDECA, EDTA plasma samples were centrifuged for 20 minutes at 2000 x g at room temperature immediately following sampling and were stored at -80°C until further analysis. For ALLVOS, EDTA plasma samples were centrifuged for 7 minutes at 2400 x g at room temperature and frozen at -80°C until further analysis within 4 hours of sampling. Protein levels were analyzed using the proximity extension assay (PEA) by Olink at SciLifeLab Affinity Proteomics Uppsala (28). In PEA, matched pairs of oligonucleotide-labeled antibodies will bind to their target antigens in a pairwise manner. Upon antibody binding, the matched oligonucleotides are brought into proximity and with the use of a DNA polymerase, a PCR target sequence is created, amplified, detected, and quantified using NGS. The Olink Explore 1536 panel used in this paper is comprised of four sub panels: Olink Explore 384 Cardiometabolic (LOT number B04413), the Explore 384 Inflammation (LOT number B04411), Olink Explore 384 Oncology (LOT number B04412)

and Explore 384 Neurology (LOT number B04414). In total, 3.7 μ l plasma per sample was used.

Following sequencing on an Illumina NovaSeq 6000, the Olink Explore platform generated raw count data where each assay/sample pair was assigned an integer value representing the number of detected DNA barcode copies. These raw counts were converted to Normalized Protein eXpression (NPX) values, which are relative quantification values on a log₂ scale and used for downstream analyses. The NPX calculation involved two main steps. First, the assay-specific counts for each sample and assay block were normalized to the corresponding Extension Control signal and subsequently log₂-transformed. The Extension Control consists of a matched antibody pair tagged with complementary DNA oligonucleotides that are always in proximity, independent of antigen binding. It provides a stable reference for controlling variation in the extension and amplification steps. Additional internal controls, including the Incubation Control and Amplification Control, were used to monitor other aspects of the assay workflow but were evaluated separately as part of quality control (QC) and not included in the actual NPX calculation.

Because samples were fully randomized across plates, it was assumed that the majority of proteins were not differentially expressed between plates. Therefore, Intensity Normalization was applied as a between-plate normalization method. In this approach the median NPX per assay across all QC-passed, non-control samples on each plate is centered to zero. This zero-centered value then serves as the reference for aligning NPX distributions across plates, effectively removing systematic intensity differences and enabling comparability across the dataset. Higher NPX equals relatively higher protein abundance, negative NPX equals lower than the reference level and a difference of one NPX means a doubling for the protein concentration. All data processing and normalization steps were performed using Olink's proprietary software package NPX Explore HT and 3072.

Additional quality control of the dataset included removing samples where more than 50% of the proteins failed quality control, excluding individual protein measurements that failed quality control, and retaining one assay for proteins measured across all four panels (TNF, IL-6, and CXCL8)."

Clarification: In total, 3.7 μ l EDTA plasma per sample was used. This since a FAST pipetting robot from Formulatrix was used. If using a Mosquito robot from SPT, only 2.8 μ l is needed which is why different amounts can be seen in different publications for this panel. As suggested, this is now in the Methods section.

Minor comments:

- Typo on line 241: "contriubuted" -> "contributed"
- Typo on line 242: "proteins proteins"

Thank you for pointing these out. We have corrected the typographical errors.

Reviewer #1 (Remarks to the Author):

The authors have provided thoughtful answers to my comments. The only issue that needs fixing is to include the scale for the OR estimates wherever it's mentioned, including in the figures. - i.e. the odds ratio per (e.g.) standard deviation increment/doubling/whatever in the xxx concentrations.

We thank the reviewer for this observation. We confirm that the reported ORs represent the effect per 1 NPX unit increase, where NPX values are expressed on a \log_2 scale. This means the OR corresponds to the effect of a doubling in relative protein concentration. We have now clarified this in the Methods section of the revised manuscript.

Reviewer #1 (Remarks to the Author)

The authors have not fairly represented the state of the MCED test space and its application to triage. "We thank the reviewer for this suggestion. Although direct comparison with commercial pan-cancer tests ... such comparisons are currently not feasible due to limited access to those assays and limited sample material." At least in the US, and I suspect in Europe, Galleri for example is commercially available, all that would be necessary is to have a clinician order it and pay for it. I accept that the material might not be available but the test is, and has been for some time. Further, although MCEds are designed to screen for early cancers, their actual training was on diagnosed cancers making it quite similar to the triage condition presented here. If an MCED can find an asymptomatic cancer, it could nearly certainly find a symptomatic cancer and it is my impression that MCEds are being used, at least occasionally, in this context.

We acknowledge the reviewer's comment. As per the editor's guidance, we have not provided additional benchmarking analyses, in line with the editorial decision that such comparisons are not required given the limited implementation of commercial pan-cancer tests in European healthcare systems.

Reviewer #3 (Remarks to the Author):

The authors have satisfactorily addressed my comments.